# Temporal variation in the diagnosis of resolved atrial fibrillation and the influence of performance targets on clinical coding: cohort study

Nicola Adderley 🅾 , Krishnarajah Nirantharakumar, Tom Marshall

Institute of Applied Health Research, University of Birmingham, Birmingham, UK

**Correspondence to**
Dr Tom Marshall;
t.p.marshall@bham.ac.uk

## ABSTRACT

**Objectives** To investigate whether the introduction of performance targets for anticoagulation in atrial fibrillation (AF) was associated with a change in use of the 'resolved AF' code.

**Design** Retrospective cohort studies.

**Setting** Data from The Health Improvement Network, a UK database of electronic patient records, from 2000 to 2016.

**Participants** 250 788 adult patients aged ≥18 years with a diagnosis of AF, including 14 757 with an incident diagnosis of 'resolved AF'.

**Main outcome measures** Annual and monthly incidence of 'resolved AF' from 2000 to 2016. Among patients with 'resolved AF', for each year we calculated median duration of the preceding AF diagnosis and the proportion prescribed anticoagulants prior to 'resolved AF'.

**Results** Incidence of 'resolved AF' increased from 5.7 to 26.3 per 1000 person-years between 2005 and the introduction of AF performance targets in 2006. Compared with the years prior to the introduction of the performance targets, incidence has remained higher in every year since their implementation. Since 2007, monthly incidence has been highest between January and March. Between 2005 and 2006, median duration between AF and 'resolved AF' diagnoses increased from 276 days (9 months) to 1343 days (3 years 8 months). Among 'resolved AF' patients with $CHA_2DS_2$-VASc score ≥1, 81.9% (95% CI 81.1 to 82.6) had no current anticoagulant prescription, and 62.3% (95% CI 61.4 to 63.2) had no record of any anticoagulant prescription.

**Conclusion** The introduction of AF performance targets was followed by a large increase in use of the 'resolved AF' code, particularly in the months immediately before practices make their anticoagulant performance target submissions. Although most AF patients are prescribed anticoagulants, few patients diagnosed with 'resolved AF' are prescribed anticoagulants and most have never been prescribed them. Untreated patients are much more likely to be coded as having 'resolved AF'.

## INTRODUCTION

Atrial fibrillation (AF) is a common cardiac arrhythmia associated with increased risk of stroke and transient ischaemic attack (TIA); this increased risk is attenuated by treatment with anticoagulants.[1–3] AF may be categorised

**Strengths and limitations of this study**

► Analysis was performed in a large primary care dataset which is generalisable to the UK population and included more than a quarter of a million patients with atrial fibrillation (AF).
► Data were derived from routinely clinical data which is used by general practitioners for clinical decision-making.
► The study explored the potential impact of the introduction of AF into the Quality and Outcomes Framework on the use of the 'resolved AF' clinical code.
► Use and interpretation of the 'resolved AF' code is likely to vary between general practitioners and practices.
► The primary care dataset contains no direct information on general practitioners' reasons for assigning a 'resolved AF' code; possible influencing factors must therefore be inferred from explorations of temporal variation, patient diagnostic information and anticoagulant prescribing.

as resolved if normal heart rhythm is restored. However, AF may recur after apparent resolution.[4 5] Evidence shows that patients diagnosed as having 'resolved AF' continue to be at increased risk of stroke/TIA; from 2013 to 2016, risk in patients with 'resolved AF' was found to be the same as that in patients with ongoing AF.[6]

Factors influencing clinicians to make a diagnosis of 'resolved AF' are unclear. Research has demonstrated that the prevalence of the AF resolved clinical code in UK general practice increased significantly after 2006 and has remained comparatively high since.[6] The Quality and Outcomes Framework (QOF) is a scheme to improve the clinical quality of care for chronic diseases. General practices keep a register of patients with particular chronic diseases and are paid an incentive for achieving performance targets for the management of patients on

the register. AF was introduced into QOF in 2006 with an incentive payment for ensuring that more than a specified percentage of patients received drugs for stroke prevention.[7] From April 2006, general practices were required to maintain a register of patients with AF and to record whether eligible patients were prescribed anticoagulants or antiplatelets; patients with a code indicating 'resolved AF' are excluded from this register. The increase in prevalence of 'resolved AF' after 2006 suggests QOF may have contributed to the increase in 'resolved AF' diagnoses. There was no corresponding jump in the recorded prevalence of AF at this time.[8] In 2012, the AF QOF indicators were updated to include an assessment of stroke risk and to require patients with a high stroke risk to be treated with anticoagulants (not antiplatelets).[9]

We hypothesised that the introduction of AF into QOF had an impact on the use of the 'resolved AF' code. The aim of this analysis, therefore, was to use information available in routinely collected primary care data to explore this hypothesis by investigating variation in the use of the 'resolved AF' clinical code over time and across different practices, and to investigate other factors which may influence general practitioners to assign a diagnosis of 'resolved AF'. The specific questions addressed were:

1. What is the annual incidence of 'resolved AF' diagnoses and did incidence increase with the introduction of AF into QOF?
2. Since the introduction of AF into QOF, is a diagnosis of 'resolved AF' more likely to be recorded in the months immediately prior to the practice QOF submission?
3. Is there a difference in the duration of AF diagnosis in patients diagnosed as having 'resolved AF' before and after the introduction of AF into QOF?
4. Are patients prescribed anticoagulants before their 'resolved AF' diagnosis?
5. How much variation exists between general practices in use of the 'resolved AF' code?

Evidence indicating that use of the 'resolved AF' code may be influenced by QOF reporting would support the recommendation that patients with 'resolved AF' be included in QOF AF registers and receive ongoing AF management,[6] or that the 'resolved AF' clinical code be withdrawn.

## METHODS
### Data source
Datasets were extracted from The Health Improvement Network (THIN), a database of electronic primary care records from UK general practices using Vision software. The version of the database from which study datasets were derived included data for approximately 14 million patients at over 640 practices. THIN comprises coded data on patient demographics, diagnoses, prescriptions issued in primary care, consultations and investigations. Data on all prescriptions issued in primary care are recorded in THIN; diagnoses that are part of the QOF are well recorded.

### Population
General practices were eligible for participation from the later of the practice acceptable mortality recording date,[10] Vision installation date plus 1 year, and the study start date (1 year prior to the first index/census date).

All adult patients aged 18 years and over with a recorded diagnosis of AF and registered for at least 365 days before the index/census date were eligible for inclusion. AF was defined by a record of a relevant clinical (Read) code.

### Study design
A retrospective cohort study from 1 January 2000 to 31 December 2016 was carried out. Index date was the latest of the following two dates: 1 year after the patient registered with the practice or the date of diagnosis of AF.

To determine incidence of 'resolved AF' among patients with AF, eligible patients were followed up from the index date until the earliest of the following: patient left the practice/transferred out, death, study end date, most recent data upload from practice, or a diagnosis of 'resolved AF'. Patients with a record of 'resolved AF' at study entry were excluded. 'Resolved AF' was defined as a record of the relevant clinical (Read) code (212R.00 'AF resolved').[6]

To explore temporal variation in AF duration and anticoagulant prescribing preceding a diagnosis of 'resolved AF', a cohort restricted to patients with a diagnosis of 'resolved AF' during the study period was used. Eligible patients were followed up until the earliest of the following: patient left practice/transferred out, death, study end date, most recent data upload from practice, or an outcome event.

To explore practice-level variation in use of the 'resolved AF' clinical code, a cross-sectional study was carried out on 1 December 2016.

### Analysis
#### Annual incidence of 'resolved AF'
Annual incidence rates of a 'resolved AF' diagnosis among AF patients were calculated for each year from 2000 to 2016 by dividing the number of patients with a new (first) record of 'resolved AF' (numerator) by the total number of person-years at risk (denominator) for the given year.

#### Monthly variation in use of the 'resolved AF' code pre-QOF and post-QOF
To investigate the impact of QOF on the distribution of 'resolved AF' coding throughout the year, monthly incidence of 'resolved AF' diagnoses (in each month from January to December) was calculated in the pre-QOF period (2000–2005), in 2006 and 2007, and in the post-QOF period (2008–2016). Monthly incidence was calculated separately for 2006 and 2007 as annual incidence of 'resolved AF' in this period, the years of and immediately following the introduction of AF into QOF, was found to be substantially higher than in subsequent years.

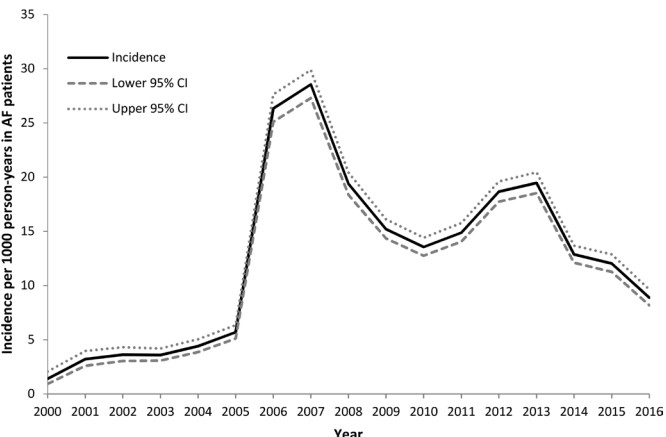

**Figure 1** Annual incidence of resolved atrial fibrillation (AF) in patients with AF 2000–2016.

In the post-QOF period (2007 onwards), Poisson regression was used to calculate crude and adjusted incidence rate ratios of stroke/TIA in patients with a 'resolved AF' diagnosis recorded in January to March compared with April to December, in order to explore any possible differences in disease severity between patients coded as resolved at different times of the year. The adjusted model included the following covariates: age, sex, $CHA_2DS_2$-VASc score (categorised as 0, 1, ≥2) and prescription of anticoagulant medication at the time of the 'resolved AF' diagnosis.[11]

### 'Resolved AF' cohort
The following analyses were restricted to patients with a record of 'resolved AF'.

#### Duration of AF diagnosis
To explore variation over time in duration of AF diagnosis in patients with 'resolved AF', median (IQR) duration of time between diagnosis of AF (earliest recorded Read code) and first record of a 'resolved AF' code was calculated for each year in patients with a 'resolved AF' code.

#### Anticoagulant prescribing
To explore prescribing of anticoagulants to patients with a diagnosis of 'resolved AF', the proportion of patients on anticoagulant treatment at the time of diagnosis (current treatment, prescribed up to 90 days prior to 'resolved AF' record), 0–90 days, and 91–180 days after the 'resolved AF' diagnosis were calculated with 95% CIs for proportions in (1) all 'resolved AF' patients and (2) patients with a $CHA_2DS_2$-VASc score ≥1 (eligible for anticoagulant treatment). The proportion of 'resolved AF' patients with a $CHA_2DS_2$-VASc score ≥1 who had never been prescribed anticoagulants was also calculated. Trends over time were explored by calculating the proportions for each year between 2000 and 2016.

### Cross-sectional analysis
#### Practice-level variation in use of 'resolved AF' code
Variation in use of the 'resolved AF' code by general practice in 2016 was assessed by plotting the percentage of AF patients with any record of a 'resolved AF' code (ever) at a given practice against the number of AF patients at the practice. Upper (UCL) and lower control limits (LCL) (within three SD of the mean) were calculated.

### Definitions of variables
AF, 'resolved AF' and stroke/TIA were defined by the presence of a clinical code; the absence of a clinical code was taken to indicate no diagnosis. The clinical code lists used have been utilised in a number of previous AF studies,[6 8 12–14] and include all codes used in QOF.[15]

$CHA_2DS_2$-VASc scores were calculated by adding 1 point each for a history of congestive heart failure (HF), hypertension, diabetes (DM), vascular disease, age 65–74 years and female sex (if another risk factor was present, otherwise 0), and 2 points for age ≥75 and a history of stroke/TIA. HF, hypertension, DM and vascular disease were defined by a relevant clinical code.

Anticoagulants included warfarin, parenteral anticoagulants, other vitamin K antagonists, and novel/non-vitamin K oral anticoagulants.

All statistical analyses were performed in Stata IC version 14.2.

### Patient involvement
Patients were not involved in the research.

## RESULTS
### Annual incidence of 'resolved AF'
A total of 250 788 patients with AF contributing 1 037 858 person-years were included in the analysis; 14 757 patients had an incident diagnosis of 'resolved AF'. Mean (SD) age was 74.6 (12.1) years; 52.6% of patients were male; median (IQR) follow-up was 3.1 (1.2–6.1) years.

Incidence of the AF resolved code in patients with AF showed a sharp rise in 2006 (figure 1), at which time AF was introduced into QOF, rising from 5.7 per 1000 person-years in 2005 to 26.3 per 1000 person-years in 2006. Incidence peaked at 28.6 per 1000 person-years in 2007; it declined thereafter, before rising again to 19.5 per 1000 person-years in 2012–2013, when further changes were made to the QOF AF requirements. Since 2013 the incidence has declined.

### Monthly variation in use of the 'resolved AF' code
Prior to the introduction of AF into QOF (January 2000 to March 2006), incidence of the 'resolved AF' code remained relatively constant across the 12 months of the year, including the 3 months immediately prior to the introduction of AF into QOF (January to March 2006), with monthly incidence varying between 3.2 and 7.2 per 1000 person-years (figure 2). From April 2006 and for the subsequent 12 months, incidence of the code steadily increased, reaching a peak of 70.2 per 1000 person-years in January 2007. From 2007 onwards (post-QOF), incidence of the 'resolved AF' code has been highest between the months of January and March, the 3 months

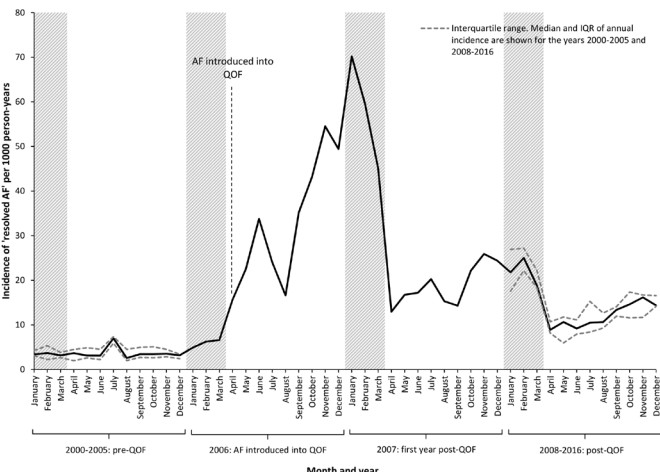

**Figure 2** Incidence of the 'resolved atrial fibrillation (AF)' code by month of recording, before, during and after the introduction of AF into the Quality and Outcomes Framework (QOF).

immediately preceding QOF report submission. In the post-QOF period (2008–2016) incidence is higher in every month of the year relative to the same month in the pre-QOF period.

From 2007 onwards, 245 patients diagnosed with 'resolved AF' in January to March and 358 patients diagnosed in April to December had a stroke. Crude incidence rates were 12.4 and 13.8 per 1000 person-years, respectively. Among patients who received a diagnosis of 'resolved AF' after the introduction of AF into QOF (2007 onwards), there was no difference in incidence of stroke/TIA in patients who were assigned the code between January and March compared with those given the code later in the year: crude incidence rate ratio (IRR) 0.90 (95% CI 0.76 to 1.06), adjusted IRR 0.98 (95% CI 0.83 to 1.15).

### 'Resolved AF' cohort

14863 patients with a record of 'resolved AF' were included in the cohort from 2000 to 2016. Median (IQR) age was 70.7 (59.6–79.6); 58.1% of patients were male. 11479 (77.2%) patients had a $CHA_2DS_2$-VASc score ≥1. Median (IQR) follow-up was 3.8 (1.9–6.8) years. 3384 (22.8%), 1737 (11.7%) and 9742 (65.5%) patients had a $CHA_2DS_2$-VASc score of 0, 1 or ≥2, respectively.

### Duration of time between diagnosis of AF and use of the 'resolved AF' code

Median duration of time between diagnosis of AF and first recording of a 'resolved AF' code remained between several months and approximately a year (varying from 69 to 335 days) between 2000 and 2005. In 2006 there was a sharp rise in median duration from 276 days (9 months) in 2005 to 1343 days (3 years 8 months) in 2006. This indicates that in 2006 more than half of patients who were assigned a 'resolved AF' code had been diagnosed over 3 ½ years earlier. Median duration then declined for

several years, before rising again to more than 1000 days in 2012–2013.

### Sequence of events in relation to anticoagulant prescribing in 'resolved AF' patients

Few patients were still on anticoagulants when the 'resolved AF' code was recorded. In the cohort of 'resolved AF' patients (2000–2016), 17.3% (95% CI 16.7 to 17.9) had a current prescription at the time of 'resolved AF' recording (up to 90 days prior), with 82.7% (95% CI 82.1 to 83.3) not being prescribed anticoagulant treatment. There was no correlation between anticoagulant prescribing and $CHA_2DS_2$-VASc category: 14.6%, 25.6% and 16.8% of patients with scores of 0, 1 and ≥2, respectively, were prescribed anticoagulants. This remained true even at high scores: among those with $CHA_2DS_2$-VASc ≥6, 14.2% were prescribed anticoagulants. Up to 90 days following the 'resolved AF' diagnosis, 9.8% (95% CI 9.3 to 10.3) of patients were still being prescribed anticoagulants. By 91 to 180 days after 'resolved AF', 8.7% (95% CI 8.3 to 9.2) had a prescription for anticoagulants.

Among 'resolved AF' patients with a $CHA_2DS_2$-VASc score ≥1, 18.1% (95% CI 17.4 to 18.9) had a current prescription for anticoagulants, while 81.9% (95% CI 81.1 to 82.6) had no current prescription. 10.5% (95% CI 10.0 to 11.1) and 9.7% (95% CI 9.2 to 10.3) had prescriptions up to 90 days and 91–180 days following the 'resolved AF' diagnosis respectively.

The proportion of 'resolved AF' patients prescribed anticoagulants shortly before recording of the 'resolved AF' code varied slightly over time, with a notable drop in 2006 to 9.8% (95% CI 8.5 to 11.4), decreasing from 25.2% (95% CI 20.6 to 30.3) in 2005.

62.3% (95% CI 61.4 to 63.2) of 'resolved AF' patients with a $CHA_2DS_2$-VASc score ≥1 had no record of an anticoagulant prescription. Among the cohort of patients whose first record of AF was after registration with the practice (n=13307), 60.6% (95% CI 59.6 to 61.5) had never been prescribed anticoagulants; this proportion varied slightly over time, reaching a peak of 70.2% in 2006 and a low of 51.3% in 2016.

### Practice-level variation

787 practices with a total of 1167771 patients with AF were included in the analysis from 2000 to 2016. 443 practices with a total of 69262 patients with AF, of whom 7261 had a record of 'resolved AF', were included in the analysis in 2016.

### Variation in use of the 'resolved AF' code between general practices

The proportion of AF patients with a record of 'resolved AF' varied between practices, ranging from 0% to 43% in 2016. The majority of practices fell within the acceptable range (between the UCL and LCL control limits) based on the size of the practice AF population, although a number of practices fell outside this range: 54 (12.2%) practices above the UCL and 30 (6.8%) below the LCL

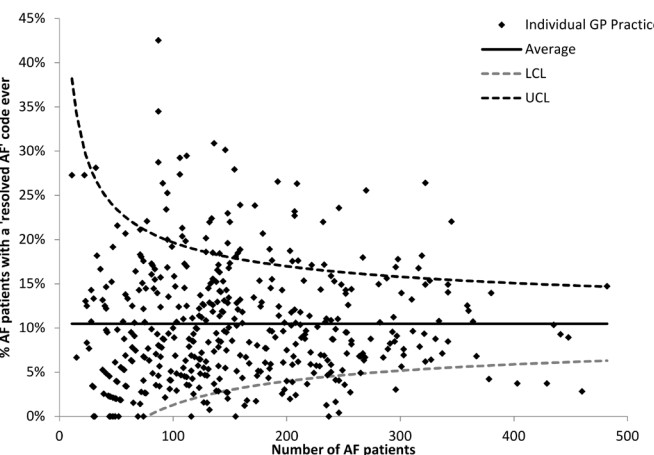

**Figure 3** Funnel plot showing variation in use of the 'resolved atrial fibrillation (AF)' code by practice in 2016.

(figure 3). In 2016, three practices with more than 100 patients with AF assigned a 'resolved AF' code to none of these patients, while 10 practices assigned a 'resolved AF' code to more than 25% of patients with AF.

Similar patterns in variation were observed in the year immediately after the introduction of AF into QOF (2007): the proportion of patients with 'resolved AF' ranged from 0% to 40%, with 61 (13.8%) practices above the UCL and 30 (6.8%) below the LCL. In 2005, immediately before the introduction of AF into QOF, there was slightly less variation: the proportion of patients with 'resolved AF' ranged from 0% to 30%, with 39 (8.8%) practices above the UCL. None were below the LCL, which was low due to the smaller average number of patients with 'resolved AF'.

## DISCUSSION

Incidence of 'resolved AF' rose dramatically in 2006 immediately following the introduction of AF into the QOF. Incidence peaked the following year at 28.6 per 1000 person-years, showing a fivefold increase compared with the incidence prior to QOF; it is possible that this increase was in part the result of practices 'catching up' with recording 'resolved AF' following the introduction of QOF. There was a further, smaller, peak in 'resolved AF' incidence in 2012–2013, following a change in the QOF AF indicators to introduce a stroke risk assessment indicator and to change the requirements for the antico-agulation indicator.[9] A corresponding rise in the prevalence of 'resolved AF' among patients with AF, from 2.3% in 2005 to 6.4% in 2007 and a high of 9.2% in 2013, has been reported previously.[6]

Since the introduction of AF into QOF, the majority of 'resolved AF' codes have been recorded between the months of January and March, immediately prior to QOF report submission by general practices. Prior to this, 'resolved AF' codes were recorded throughout the year with little monthly variation in incidence. There is no difference in stroke/TIA rates in patients diagnosed

as having 'resolved AF' between January and March compared with those diagnosed later in the year; patients with AF who are diagnosed as resolved immediately prior to QOF do not have a different/lower risk of stroke/TIA.

Immediately following the introduction of AF into QOF, there was a dramatic rise in median duration between AF and 'resolved AF' diagnoses, with a further peak at the time of changes to QOF in 2012–2013. At these time points, patients designated as having 'resolved AF' had been diagnosed with AF several years previously (median 3 years and 8 months in 2006) compared around 1 year prior to QOF (9 months in 2005).

Almost two-thirds of patients with 'resolved AF' and a $CHA_2DS_2$-VASc score ≥1 had never been prescribed anti-coagulants. In 2016, 79.5% of patients with 'resolved AF' and a $CHA_2DS_2$-VASc score ≥1 were not prescribed anti-coagulants at the time of their 'resolved AF' diagnosis, made up of 53.5% who had never been prescribed antico-agulants and 26.0% who had previously been prescribed anticoagulants but had subsequently discontinued. By contrast, only 25%–30% of patients with ongoing AF and a $CHA_2DS_2$-VASc score ≥1 were not prescribed anti-coagulants in 2016.[8 16] This suggests that patients with AF who are not prescribed anticoagulants may be more likely to be assigned a 'resolved AF' code. Furthermore, recent evidence indicates that patients with a diagnosis of 'resolved AF' remain at increased risk of stroke/TIA and may therefore benefit from continued anticoagulant prophylaxis.[6] The concept of 'resolved AF' may be delu-sive; AF which has apparently resolved, even following ablation, may recur.[17–19]

Use of the 'resolved AF' code varies between practices. Some practices with large numbers of AF patients use the code for very few patients, while others assign the code to more than a quarter of AF patients.

### Strengths and limitations

This analysis was performed in a large general practice dataset which is generalisable to the UK population. Data were derived from routinely collected clinical data which is used by general practitioners for clinical decision-making. The use and interpretation of the 'resolved AF' clinical code is likely to vary between general practi-tioners and practices. The primary care dataset contains no direct information on general practitioners' reasons for assigning a 'resolved AF' code; possible influencing factors have therefore been inferred from explorations of temporal variation, patient diagnostic information and anticoagulant prescribing. In order to better understand the factors motivating a diagnosis of 'resolved AF', a qual-itative study and consultation with practicing clinicians would be required.

Anticoagulation rates may be underestimated if treat-ment is managed entirely in secondary care; however, the majority of anticoagulants are prescribed in primary care. AF clinical guidelines and stroke risk scoring systems have changed over the study period; for the purpose of this study, we used current guidance (eligibility for

anticoagulation based on $CHA_2DS_2$-VASc score $\geq$1) across all time periods for consistency and comparability.

## Conclusions

Use of the 'resolved AF' code remains common. Most patients eligible for anticoagulant treatment who were assigned a 'resolved AF' code were never prescribed anticoagulants, and very few patients were still taking anticoagulants when the 'resolved AF' code was recorded. Those diagnosed as having 'resolved AF' are no longer included in the AF register for QOF; this has the effect of improving the practice's apparent performance in the QOF. Incidence of the 'resolved AF' clinical code increased markedly when AF was introduced into QOF in 2006 and increased again when further changes were made to the QOF incentive scheme in 2012. Since 2006, incidence of the 'resolved AF' code has been highest in the months shortly before practices make their QOF submissions. Previous evidence demonstrated patients with a diagnosis of 'resolved AF' remain at increased risk of stroke/TIA and are therefore likely to benefit from anticoagulant prophylaxis. We therefore recommend that patients with 'resolved AF' should be included when determining whether practices meet QOF clinical performance targets.

**Contributors** NA, KN and TM designed the study. KN undertook data extraction. NA designed and performed the analyses. NA wrote the first draft of the paper, which was revised in collaboration with TM and KN. NA acts as guarantor.

**Funding** This research (NA and TM) was supported by the National Institute for Health Research Collaboration for Leadership in Applied Health Research and Care West Midlands (NIHR CLAHRC WM), now recommissioned as NIHR Applied Research Collaboration West Midlands. The views expressed in this publication are those of the author(s) and not necessarily those of the NIHR or the Department of Health and Social Care. The funders had no role in study design, data collection and analysis, decision to publish, or preparation of the manuscript.

**Competing interests** All authors have completed the ICMJE uniform disclosure form at http://www.icmje.org/coi_disclosure.pdf and declare: NA and TM report a grant from the National Institute for Health Research (NIHR) Collaboration for Leadership in Applied Health Research and Care West Midlands during the conduct of the study; KN reports funding from AstraZeneca and fees from Sanofi and Boehringer Ingelheim outside the submitted work. Authors declare no other financial relationships with any organisations that might have an interest in the submitted work in the previous three years; and no other relationships or activities that could appear to have influenced the submitted work.

**Patient consent for publication** Not required.

**Ethics approval** The THIN data collection scheme and research carried out using THIN data were approved by the NHS South-East Multicentre Research Ethics Committee (MREC) in 2003; under the terms of this approval, studies must undergo independent scientific review. Approval for these analyses was obtained from the Scientific Review Committee (for the use of THIN data) in April 2015 (15THIN021) and September 2017 (SRC reference number 17THIN082).

**Provenance and peer review** Not commissioned; externally peer reviewed.

**Data availability statement** No data are available.

**ORCID iD**
Nicola Adderley http://orcid.org/0000-0003-0543-3254

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
