## [Reviewer comments · BMJ Open]

ARTICLE DETAILS

TITLE (PROVISIONAL)	Temporal variation in the diagnosis of resolved atrial fibrillation and the influence of performance targets on clinical coding: cohort study
AUTHORS	Adderley, Nicola; Nirantharakumar, Krishnarajah; Marshall, Tom

VERSION 1 – REVIEW

REVIEWER	Ming-Hsiung Hsieh Wan-Fang Hospital, Taipei Medical University, Taipei, Taiwan
REVIEW RETURNED	29-Apr-2019

GENERAL COMMENTS	The authors investigate an interesting clinical issue regarding "revolved AF" in real world clinical practice. They found that most of these patients did not receive anticoagulants even they had higher CHA2DS2VASc score. This clinical decision will raise the higher risk of ischemic stroke. Because of retrospective study, the authors did not study why those patients became revolved AF. In addition, there is a big variation in different regions. Only one minor suggestion regarding this manuscript: please shorten the "conclusion" to only one paragraph.
---

REVIEWER	Gaurav A Upadhyay University of Chicago, Chicago IL USA
REVIEW RETURNED	05-May-2019

GENERAL COMMENTS	Adderley, Nirantharakumar, and Marshall present their analysis of the use of the "resolved AF" diagnosis utilizing The Health Improvement Network (THIN) between 2000 and 2016. This builds on prior work by the same group regarding the risk of CVA. They found that incidence of the resolved AF diagnosis increased significantly after introduction of AF performance targets in 2006. They found that the monthly incidence of the term was greater between January and March, prior to Quality and Outcomes Framework (QOF) submission. Importantly, they also note that use of anticoagulants were low in patients with the diagnosis, despite CHADS2-VASc score ≥ 1 . They suggest that this discrepancy is to improve apparent performance based on removal from the AF register, although this conclusion is largely inferential. This is important work for which the authors should be commended. With that said, in its present form the work does extend somewhat beyond the data at hand regarding the motivations behind changes in coding and could benefit from some additional analyses to strengthen their claims. Further comments and questions are elaborated on below: - It is notable that the longest section of the abstract is the conclusion. Would suggest amplifying the Results section and trimming some of the Conclusions section, particularly regarding
---

	inferences regarding the motivation of general practices, which cannot be gleaned from these data  - Please expand on the implications of the QOF in the Introduction which will not be apparent to your international readers. What are the implications of having lower rates of anticoagulation post-QOF? Is there further inquiry into practices or patients? Or is it simply a nuisance to participate? - What was the mean or median time that a patient remained in the THIN database? - How complete are records regarding anticoagulant prescribing? - Building upon your prior work, were patients who were not prescribed an anticoagulant with the “resolved AF” code more likely to have a stroke than similarly matched patients with the AF code who were on anticoagulation. This is likely the case, but if a subset could be analyzed in a case-control fashion, this would support the conclusions substantially - Could the spike in use of the resolved AF in 2006 simply represent a “catch-up” period for patients in whom this finding had been present but was simply not coded? It appears that there was a relative plateau in the incidence at least after the first year based on review of Figure 2. - Since rate of stroke/TIA is the same for patients given the diagnosis in Jan-March versus later in the year, why might some patients be given the designation earlier? Could it simply reflect presentation time to the general practices - Please provide the distribution of CHADS2-VASc scores in the resolved AF group. Is this similar or different than those with AF more broadly? Is there any association with anticoagulation and the score? - I am surprised that patients with “resolved AF” with high CHADS2-VASc scores were never treated with anticoagulants. Is that still the case when looking at scores >3? - Patients with “resolved AF” may have had postoperative AF or paroxysmal AF in the setting of another illness. Is there a way to get insight into the presence of other comorbid states or procedures for these patients?
--	--

REVIEWER	James Brophy McGill University
REVIEW RETURNED	13-Jul-2019

GENERAL COMMENTS	Title: Temporal variation in the diagnosis of resolved atrial fibrillation and the influence of performance targets on clinical coding: cohort study Summary: Cohort study investigating how the introduction of performance targets for anticoagulation in atrial fibrillation (AF) affected use of the ‘resolved atrial fibrillation’ code. General comments: Although I am a cardiologist, I was asked to concentrate my review on the statistical aspects of the submission. Nevertheless, I do have two clinical questions; 1. The authors state “risk in patients with ‘resolved AF’ was found to be the same as that in patients with ongoing AF.6” Reference 6 is previous work from the same authors and the abstract of that paper states “Adjusted incidence rate ratios for stroke or TIA in patients with resolved atrial fibrillation were 0.76 (95% confidence interval 0.67 to 0.85, P<0.001) versus controls with atrial fibrillation”. Perhaps I have misunderstood but this appears to be a direct
--

	contradiction. 2. I don't know the history of CHADS-Vasc guidelines in the UK but in Canada, initially anticoagulation was recommended only for those with scores > 2. In or around 2012 the guidelines for anticoagulation, with no additional evidence that I recall, were changed to ≥ 1. If the same situation existed in the UK using the guideline of ≥ 1 for the whole period would be potentially misleading. On a methodological level, I also have several comments. 1. Did the authors have a prespecified research protocol that a priori listed their objectives that could be referenced? I am impressed that the authors would be so prescient as to believe that the overdiagnosis of resolved a fib would occur only in the 3 months before April reporting, exactly as what was observed in the data. 2. The authors' conclusion "This suggests general practices are choosing to code some patients as having 'resolved AF', thereby removing these patients from the AF register, in order to improve their apparent performance." over reaches the causal inferences possible from this observational study design. Regarding statistics, the authors state that adjusted Poisson regression was used. The exact model regression coefficients are not presented and there is no mention of assessing the degree of model fit. Looking at Figure 1, it is hard to imagine how Poisson regression, a log linear type model, can adequately model this data. I would suggest that consideration be given to a statistical consultation and exploration of a possible time series analysis. The authors also evaluate practice level variation. The data within practices are undoubtedly not independent but clustered in a hierarchical manner. Unless this hierarchy is considered standard errors will be incorrect as will be associated statistical inferences.
--	--

VERSION 1 – AUTHOR RESPONSE

Reviewer 1 comments:

Only one minor suggestion regarding this manuscript: please shorten the "conclusion" to only one paragraph.

As requested, the conclusion has been shortened to one paragraph.

Reviewer 2 comments:

This is important work for which the authors should be commended. With that said, in its present form the work does extend somewhat beyond the data at hand regarding the motivations behind changes in coding and could benefit from some additional analyses to strengthen their claims. Further comments and questions are elaborated on below:

We thank the reviewer for the positive feedback.

It is notable that the longest section of the abstract is the conclusion. Would suggest amplifying the Results section and trimming some of the Conclusions section, particularly regarding inferences regarding the motivation of general practices, which cannot be gleaned from these data

As suggested by the reviewer, the conclusions section of the Abstract has been shortened, and the results slightly expanded to include additional information on incidence trends and data on duration between AF and 'resolved AF' diagnoses.

Please expand on the implications of the QOF in the Introduction which will not be apparent to your international readers. What are the implications of having lower rates of anticoagulation post-QOF? Is there further inquiry into practices or patients? Or is it simply a nuisance to participate?

We have now added the following information about QOF to the introduction:

The Quality and Outcomes Framework (QOF) is a scheme to improve the clinical quality of care for chronic diseases. General practices keep a register of patients with particular chronic diseases and are paid an incentive for achieving performance targets for the management of patients on the register. AF was introduced into QOF in 2006 with an incentive payment for ensuring that more than a specified percentage of patients received drugs for stroke prevention.

What was the mean or median time that a patient remained in the THIN database?

Median (IQR) follow-up period for the cohort study was 3.8 (1.9-6.8) years (study entry date was the maximum of the date of 'resolved AF' diagnosis and one year after practice registration). This information has now been added to the Results section.

How complete are records regarding anticoagulant prescribing?

Prescribing data are generally complete in general practice records.¹ All prescriptions issued in primary care are captured in THIN. Prescriptions made in secondary care may not be recorded, but the majority of anticoagulants are prescribed in primary care. The following has been added to the Methods and Strength and limitations sections:

Methods, Data source:

Data on all prescriptions issued in primary care are recorded in THIN; diagnoses that are part of the QOF are well recorded.

Strengths and limitations:

Anticoagulation rates may be underestimated if treatment is managed entirely in secondary care; however, the majority of anticoagulants are prescribed in primary care.

Building upon your prior work, were patients who were not prescribed an anticoagulant with the "resolved AF" code more likely to have a stroke than similarly matched patients with the AF code who were on anticoagulation. This is likely the case, but if a subset could be analyzed in a case-control fashion, this would support the conclusions substantially

At the reviewer's request, we have compared stroke/TIA incidence in 'resolved AF' patients who were and were not prescribed anticoagulants. Stroke/TIA rates were 25% higher in 'resolved AF' patients without a current anticoagulant prescription (at 'resolved AF' diagnosis date) compared to those with a prescription: adjusted IRR 1.25 (95% CI 1.02 to

1.53). Among 'resolved AF' patients who had no anticoagulant prescription within 90 days after the 'resolved AF' diagnosis, stroke/TIA incidence was 40% higher than in those with a

¹ Whitelaw FG, Nevin SL, Milne RM, Taylor RJ, Taylor MW, Watt AH. Completeness and accuracy of morbidity and repeat prescribing records held on general practice computers in Scotland. Br J Gen Pract 1996; 46: 181-186.

continuing prescription: adjusted IRR 1.40 (95% CI 1.06 to 1.84). However, as noted, prescribing rates are very low among 'resolved AF' patients, leading to small numbers of patients in the prescribed groups and wide confidence intervals.

Could the spike in use of the resolved AF in 2006 simply represent a "catch-up" period for patients in whom this finding had been present but was simply not coded? It appears that

there was a relative plateau in the incidence at least after the first year based on review of Figure 2.

It is possible that the spike in 2006 was partially due to a catch-up in documenting 'resolved AF'; however, incidence of 'resolved AF' has remained consistently higher in the years since QOF compared to the years prior to QOF.

Since rate of stroke/TIA is the same for patients given the diagnosis in Jan-March versus later in the year, why might some patients be given the designation earlier? Could it simply reflect presentation time to the general practices

We reported no difference in the risk of stroke/TIA between patients diagnosed as 'resolved AF' in the January to March period and those diagnosed in other months. We believe the fact that stroke risk is similar in patients diagnosed at different times of the year suggests that these patients do not differ clinically or in terms of AF severity. We suspect that increased rate of diagnosis of 'resolved AF' in January to March is linked to the QOF submission date at the start of April. However, we cannot determine from the dataset why clinicians diagnose 'resolved AF' at any particular time of the year. It is possible that it is related to when patients consult.

Please provide the distribution of CHADS2-VASc scores in the resolved AF group. Is this similar or different than those with AF more broadly? Is there any association with anticoagulation and the score?

Among the 'resolved AF' cohort, 3384 (22.8%), 1737 (11.7%) and 9742 (65.5%) had a CHA2DS2-VASc score of 0, 1 or ≥ 2 , respectively. We previously found CHA2DS2-VASc scores were slightly higher in patients with AF compared to 'resolved AF'; this is reported in our earlier paper (Adderley, Nirantharakumar and Marshall, BMJ 2018;361:k1717): mean (SD) CHA2DS2-VASc scores were 2.5 (1.9) and 3.7 (1.7) in 'resolved AF' and age- and sex-matched AF patients, respectively.

There was no correlation between anticoagulant prescribing rates at index date and

CHA2DS2-VASc category: 14.6%, 25.6% and 16.8% of 'resolved AF' patients with scores of 0, 1 and ≥ 2 , respectively, were prescribed anticoagulants. This information has been added to the results.

I am surprised that patients with "resolved AF" with high CHADS2-VASc scores were never treated with anticoagulants. Is that still the case when looking at scores >3?

As noted above, there was no clear correlation between prescribing rates and CHA2DS2-VASc score. However, at the reviewer's request, we have looked at the prescribing rate in patients with a score of 3 or more: 15.6% had a current prescription at index date.

Patients with "resolved AF" may have had postoperative AF or paroxysmal AF in the setting of another illness. Is there a way to get insight into the presence of other comorbid states or procedures for these patients?

Possible diagnoses underlying the coding of 'resolved AF' were explored in our earlier paper (BMJ 2018;361:k1717), including assessing stroke risk in 'resolved AF' patients who previously had a record of paroxysmal AF or a record of ablation. We found that the pattern of comorbidities was similar between patients with 'resolved AF' age- and sex-matched patients with AF, although prevalence of comorbidities was slightly lower in 'resolved AF' (cf. Table 1, BMJ 2018;361:k1717). However, the presence of other comorbidities does not explain observed changes in use of the 'resolved AF' diagnosis following the introduction of performance targets.

Reviewer 3 comments:

The authors state "risk in patients with 'resolved AF' was found to be the same as that in patients with ongoing AF.6 ' Reference 6 is previous work from the same authors and the abstract of that paper states "Adjusted incidence rate ratios for stroke or TIA in patients with resolved atrial fibrillation were 0.76 (95% confidence interval 0.67 to 0.85, P<0.001)

versus controls with atrial fibrillation". Perhaps I have misunderstood but this appears to be a direct contradiction.

The reviewer is correct that the previous work found that overall risk of stroke/TIA in patients with 'resolved AF' was lower than in those with ongoing AF over the whole study period (2000-2016). However, when we undertook a subgroup analysis stratified by year of diagnosis of 'resolved AF', there was a clear trend towards a smaller difference over time and in the most recent time periods, the difference between ongoing AF and 'resolved AF' was not statistically significant: adjusted IRR 0.81 (95% CI 0.65 to 1.00), 0.79 (0.61 to 1.02), and 0.96 (0.67 to 1.39) for 2007-10, 2010-13 and 2013-16, respectively (see 'Temporal trends' section of Results in Adderley, Nirantharakumar and Marshall, BMJ 2018;361:k1717). As noted in the Introduction of our submitted manuscript, for 2013-16, we found no difference in risk of stroke/TIA between patients with 'resolved AF' and ongoing AF.

I don't know the history of CHADS-Vasc guidelines in the UK but in Canada, initially anticoagulation was recommended only for those with scores > 2. In or around 2012 the guidelines for anticoagulation, with no additional evidence that I recall, were changed to >=1. If the same situation existed in the UK using the guideline of >=1 for the whole period would be potentially misleading.

Since the introduction of AF into the Quality and Outcomes framework in 2006, UK NICE guidelines have recommended anticoagulant therapy for patients with high stroke risk

(initially according to their own algorithm, and then using CHADS2 and subsequently CHA2DS2-VASc score ≥ 2), and recommended anticoagulants or antiplatelets for those with moderate risk (CHA2DS2 or CHA2DS2-VASc score ≥ 1).^{2,3,4} Therefore, patients with moderate or high stroke risk were eligible for anticoagulant treatment. We utilised CHA2DS2-VASc score throughout the study period for consistency and comparability, and included all patients eligible for anticoagulant treatment (i.e. moderate to high risk, with a score ≥ 1) in the denominator. This is consistent with the definition we have previously used (BMJ 2018;361:k1717; Heart 2019;105:27-33).

On a methodological level, I also have several comments.

- 1. Did the authors have a prespecified research protocol that a priori listed their objectives that could be referenced? I am impressed that the authors would be so prescient as to believe that the overdiagnosis of resolved a fib would occur only in the 3 months before April reporting, exactly as what was observed in the data.*

There was no pre-specified research protocol; this research evolved from the findings presented in our previous work, which indicated that prevalence of 'resolved AF' increased after the introduction of AF into performance targets (QOF). The reason for undertaking monthly analysis was because we suspected that, if there were an association between QOF and use of the diagnostic code, the rates of 'resolved AF' might be higher in the month prior to the April QOF submission date.

- 2. The authors' conclusion "This suggests general practices are choosing to code some patients as having 'resolved AF', thereby removing these patients from the AF register, in order to improve their apparent performance." over reaches the causal inferences possible from this observational study design.*

We agree with the reviewer that our original conclusion overstated what can be inferred from the evidence presented. We have now re-written the conclusion and removed this assertion. The conclusion now reads as follows:

Use of the 'resolved AF' code remains common. Most patients eligible for anticoagulant treatment who were assigned a 'resolved AF' code were never prescribed anticoagulants, and very few patients were still taking anticoagulants when the 'resolved AF' code was recorded. Those diagnosed as having 'resolved AF' are no longer included in the AF register for QOF; this has the effect of improving the practice's apparent performance in the QOF. Incidence of the 'resolved AF' clinical code increased markedly when AF was introduced into QOF in 2006 and increased again when further changes were made to the QOF incentive scheme in 2012. Since 2006, incidence of the 'resolved AF' code has been highest in the months shortly before practices make their QOF submissions. Previous

² National Institute for Health and Clinical Excellence. Atrial fibrillation. The management of atrial fibrillation. NICE Clinical Guideline 36. London: National Institute for Health and Clinical Excellence 2006.

³ National Institute for Health and Care Excellence. Atrial fibrillation: the management of atrial fibrillation. NICE clinical guideline 180. National Institute for Health and Care Excellence 2014.

⁴ NHS Employers and General Practitioners Committee. Quality and Outcomes Framework for 2012/13. Guidance for PCOs and practices. NHS Employers: London 2012.

evidence demonstrated patients with a diagnosis of ‘resolved AF’ remain at increased risk of stroke/TIA and are therefore likely to benefit from anticoagulant prophylaxis. We therefore recommend that patients with ‘resolved AF’ should be included when determining whether practices meet QOF clinical performance targets.

Regarding statistics, the authors state that adjusted Poisson regression was used. The exact model regression coefficients are not presented and there is no mention of assessing the degree of model fit.

As described in the Methods section, a Poisson regression model was only used to calculate adjusted incidence rate ratios for stroke/TIA. We were not producing a prediction model, and therefore reporting model coefficients and measures of model fit/performance would not be standard practice. We have followed standard guidance for reporting adjusted regression model methods and results, including reporting the covariates included in the model along with crude and adjusted IRR in the results. Please see also the response to your following query.

Looking at Figure 1, it is hard to imagine how Poisson regression, a log linear type model, can adequately model this data. I would suggest that consideration be given to a statistical consultation and exploration of a possible time series analysis.

Poisson regression was not used to model the data presented in Figure 1, which shows trends in ‘resolved AF’ incidence over time. For incidence over time, we reported and plotted only crude incidence to illustrate changes over time. There was no significant change in the underlying population demographics of the THIN database used over this period.

The authors also evaluate practice level variation. The data within practices are undoubtedly not independent but clustered in a hierarchical manner. Unless this hierarchy is considered standard errors will be incorrect as will be associated statistical inferences.

The practice-level data we present are purely descriptive: we present only the percentage of AF patients assigned the ‘resolved AF’ code within individual practices in order to describe the extent to which use of the ‘resolved AF’ code varies between practices; we make no further statistical inferences. Therefore, we do not believe these results will be impacted by any possible within-practice data clustering.

VERSION 2 – REVIEW

REVIEWER	Gaurav A. Upadhyay University of Chicago Medicine
REVIEW RETURNED	20-Aug-2019
GENERAL COMMENTS	Adderley and colleagues present here a revision of their manuscript on the use of the “resolved AF” diagnosis utilizing The Health Improvement Network (THIN) between 2000 and 2016. This revision is substantially improved from the initial submission, particularly with respect to focusing discussion on the available data. Importantly, the conclusions that use of the term increased in 2006 and again in

	2012 are nicely supported by the present data. Importantly, they also comment on the overall low use of anticoagulants in patients with CHADS2-VASc score ≥ 1. The authors should be commended for this valuable work which raises awareness of an important possible pitfall in the current coding/QOF guidelines. Some additional comments below:  - Would note that the initial increase in the use of the “resolved AF” code in 2006 may have been due to “catch up” because of change in the QOF guidelines themselves rather than over or miscoding - Importantly, from a physiologic perspective, there are rare—if any situations—in which AF can ever truly be “resolved.” It is increasingly being recognized that even postoperative AF, previously thought to be relatively benign, is associated with increased risk comparable that to paroxysmal AF (see Butt et al., Risk of Thromboembolism Associated With Atrial Fibrillation Following Noncardiac Surgery. Journal of the American College of Cardiology. Oct 23 2018;72(17):2027-2036). Would consider alluding to this in the discussion as well while raising question with use of the diagnosis “resolved AF” which may be a misnomer itself - Were patients with very high CHADS2-VASc scores in the “resolved AF” code also not anticoagulated? What proportion of these patients? Presumably these patients would be at the highest risk for TIA and is worth flagging. - Were patients with the “resolved AF” code fundamentally different than patients with paroxysmal or persistent AF? If they demonstrated greater comorbidities associated with the risk of bleeding, this may also explain why some practitioners would choose to use this term rather than anticoagulate, although this remains speculative. If it is possible to compare demographics of patients with “resolved AF” versus paroxysmal or persistent AF, that may provide greater insight into use of the code and would strengthen the Discussion
--	---

REVIEWER	James Brophy McGill University CANADA
REVIEW RETURNED	04-Aug-2019

GENERAL COMMENTS	Title: Temporal variation in the diagnosis of resolved atrial fibrillation and the influence of performance targets on clinical coding: cohort study General comments: This is a resubmission of a manuscript that I previously reviewed. I appreciate the authors’ responses to the earlier comments even if I feel that have only partially addressed my concerns. 1. The manuscript still contains causal language that I don’t think is supported by the study design – cf i) In the abstract – “target performance... AFFECTED use of the ‘resolved atrial fibrillation’ ii) The study assessed the impact of the introduction of AF into the Quality and Outcomes Framework on the use of the ‘resolved AF’ clinical code. Iii) ‘resolved AF’ code is substantially driven by QOF 2. The notion that anticoagulation was the standard recommended therapy throughout the study period is simply not supported by the NICE 2006 guidelines that they refer to. A summary of those guidelines published in Heart in Jan 2007 contains the attached figure Note that it is stated that with 2 or more RF anticoagulation maybe
--

	preferred. This confirms my Canadian experience that at least until 2012, ASA was judged to be an acceptable recommended treatment for CHADS =1, yet the authors code CHADS =>1 as not meeting the recommendations. This does not seem to reflect the contemporary NICE guidelines, at least until 2012 and will exaggerate their results. Consequently, patients with resolved a fib and CHADS =1 may quite appropriately not have received anticoagulation. 3. The researchers have played with several degrees of freedom that will not obvious to the casual reader and are not acknowledged in the manuscript. i) They found that resolved a fib had a lower stroke / TIA rate in their study (2000-2016) but elect only to refer to a subgroup period of 2013-16 in the present study. i)They claim to be especially interested in the period Jan-Mar as if this was prespecified, which apparently it wasn't judging from their previous response. Further there is no statistical testing of the observation that rates are highest in the first 3 months. Also Figure 2 could present the spread of the data for the combined 2008-2016 period. I do not feel that these comments represent "fatal flaws" but simply represent honest and acceptable scientific differences of opinion.
--	--

VERSION 2 – AUTHOR RESPONSE

Reviewer 2 Comments:

The authors should be commended for this valuable work which raises awareness of an important possible pitfall in the current coding/QOF guidelines.

We thank the reviewer for the positive feedback.

Would note that the initial increase in the use of the "resolved AF" code in 2006 may have been due to "catch up" because of change in the QOF guidelines themselves rather than over or miscoding

As recommended, we have now added the following to the discussion: 'it is possible that this increase was in part the result of practices 'catching up' with recording 'resolved AF' following the introduction of QOF'.

Importantly, from a physiologic perspective, there are rare—if any situations—in which AF can ever truly be "resolved." It is increasingly being recognized that even postoperative AF, previously thought to be relatively benign, is associated with increased risk comparable that to paroxysmal AF (see Butt et al., Risk of Thromboembolism Associated With Atrial Fibrillation Following Noncardiac Surgery. Journal of the American College of Cardiology. Oct 23 2018;72(17):2027-2036). Would consider alluding to this in the discussion as well while raising question with use of the diagnosis "resolved AF" which may be a misnomer itself

We fully agree with the reviewer that 'resolved AF' is likely to be a misnomer. This was briefly alluded to in our earlier BMJ paper; however, we have now added the following to the

discussion: 'The concept of 'resolved AF' may be delusive; AF which has apparently resolved, even following ablation, may recur.^{1,2,3'}

Were patients with very high CHADS2-VASc scores in the "resolved AF" code also not anticoagulated? What proportion of these patients? Presumably these patients would be at the highest risk for TIA and is worth flagging.

The table and chart below show the number and percentage of patients with 'resolved AF' with each CHA2DS2-VASc score prescribed/not prescribed anticoagulants.

CHADS-VASc score	On anticoagulant		Not on anticoagulant		Total
	n	%	n	%	
0	493	14.6	2,891	85.4	3,384
1	445	25.6	1,292	74.4	1,737
2	551	19.8	2,237	80.2	2,788
3	487	16.9	2,392	83.1	2,879
4	359	14.8	2,069	85.2	2,428
5	151	14.9	860	85.1	1,011
6	68	14.9	389	85.1	457
7	14	9.5	133	90.5	147
8	7	28.0	18	72.0	25
9	1	14.3	6	85.7	7
Total	2,576	17.3	12,287	82.7	14,863

¹ Chao TF, Lin YJ, Chang SL, et al. Can oral anticoagulants be stopped safely after a successful atrial fibrillation ablation? *J Thorac Dis* 2015;7:172-7. doi: 10.3978/j.issn.2072-1439.2015.01.18

² Chao TF, Tsao HM, Lin YJ, et al. Clinical outcome of catheter ablation in patients with nonparoxysmal atrial fibrillation: results of 3-year follow-up. *Circ Arrhythm Electrophysiol* 2012;5:514-20.

³ Tilz RR, Rillig A, Thum AM, et al. Catheter ablation of long-standing persistent atrial fibrillation: 5-year outcomes of the Hamburg Sequential Ablation Strategy. *J Am Coll Cardiol* 2012;60:1921-9.

As previously noted, there is no correlation between CHA2DS2-VASc score and prescribing rates. We have now expanded the results section to note: 'This remained true even at high scores: among those with CHA2DS2-VASc ≥ 6 , 14.2% were prescribed anticoagulants.'

Were patients with the "resolved AF" code fundamentally different than patients with paroxysmal or persistent AF? If they demonstrated greater comorbidities associated with the risk of bleeding, this may also explain why some practitioners would choose to use this term rather than anticoagulate, although this remains speculative. If it is possible to compare demographics of patients with "resolved AF" versus paroxysmal or persistent AF, that may provide greater insight into use of the code and would strengthen the Discussion

Unfortunately, comparing demographics and comorbidities of 'resolved AF' patients with patients with paroxysmal or persistent AF was beyond the scope of this study, as we do not have this data in our study dataset. However, in a previous study carried out by our team looking at stroke risk in paroxysmal AF, we obtained data on contraindications to anticoagulants; therefore, for the reviewer's reference, we have used this data to summarise information on mean age and presence of contraindications to anticoagulants for patients with paroxysmal AF and 'resolved AF' in the table below (data for 2015):

	Paroxysmal AF	Resolved AF
Population	57453	7876
Age (mean)	76.3	70.3
Contraindications, n (%)		
Adverse reaction	68 (0.1)	6 (0.1)
Aneurysm	1525 (2.7)	185 (2.3)
Bleed	12073 (21.0)	1528 (19.4)
Haemorrhagic stroke	940 (1.6)	103 (1.3)
Oesophageal varices	73 (0.1)	15 (0.2)
Retinopathy	111 (0.2)	12 (0.2)
Ulcer	3408 (5.9)	474 (6.0)

While the age of paroxysmal AF patients was slightly higher than 'resolved AF' patients in this dataset, the distribution of these particular comorbidities/contraindications was very similar across the two groups.

Reviewer 3 Comments:

1. *The manuscript still contains causal language that I don't think is supported by the study design – cf i) In the abstract – “target performance... AFFECTED use of the ‘resolved atrial fibrillation’ ii) The study assessed the impact of the introduction of AF into the Quality and Outcomes Framework on the use of the ‘resolved AF’ clinical code. lii) ‘resolved AF’ code is substantially driven by QOF*

Thank you for noting these language inconsistencies. We have now amended each as follows:

3. Abstract: ‘To investigate whether the introduction of performance targets for anticoagulation in atrial fibrillation (AF) was associated with a change in use of the ‘resolved atrial fibrillation’ code.’

4. ‘The study explored the potential impact of the introduction of AF into the Quality and Outcomes Framework on the use of the ‘resolved AF’ clinical code.’

5. ‘Evidence indicating that use of the ‘resolved AF’ code may be influenced by QOF reporting...’

The notion that anticoagulation was the standard recommended therapy throughout the study period is simply not supported by the NICE 2006 guidelines that they refer to. A summary of those guidelines published in Heart in Jan 2007 contains the attached figure Note that it is stated that with 2 or more RF anticoagulation maybe preferred. This confirms my Canadian experience that at least until 2012, ASA was judged to be an acceptable recommended treatment for CHADS =1, yet the authors code CHADS =>1 as not meeting the recommendations. This does not seem to reflect the contemporary NICE guidelines, at least until 2012 and will exaggerate their results. Consequently, patients with resolved a fib and CHADS =1 may quite appropriately not have received anticoagulation.

Although there was evidence for the effectiveness of anticoagulants before 2006 there was uncertainty that anticoagulants were more effective than antiplatelet agents until 2007.^{4,5,6} We acknowledge that anticoagulants were the recommended treatment for patients with a CHADS2 score of 2 or more; however, patients with a score of 1 were also eligible for anticoagulants, as indicated in the central path of the guidelines/figure (‘consider anticoagulation or aspirin’). This is the terminology/language we have employed throughout the manuscript: we describe the proportion of eligible patients who were prescribed anticoagulants – those for whom anticoagulants were a recommended treatment option, not necessarily those for whom this was the only recommended treatment.

We also recognise that treatment recommendations and stroke risk scoring systems have changed over the study period; we chose to use a consistent way of presenting the data (patients eligible for anticoagulants based on the CHA2DS2-VASc score) throughout all time periods for consistency and comparability. We have now acknowledged this in the Strengths and limitations section of the manuscript, as follows: ‘AF clinical guidelines and stroke risk scoring systems have changed over the study period; for the purpose of this study, we used current guidance (eligibility for anticoagulation based on CHA2DS2-VASc score \geq 1) across all time periods for consistency and comparability.’

- 2 *The researchers have played with several degrees of freedom that will not obvious to the casual reader and are not acknowledged in the manuscript. i) They found that resolved a fib had a lower stroke / TIA rate in their study (2000-2016) but elect only to refer to a*

⁴ Benavente O, Hart R, Koudstaal P, Laupacis A, McBride R. Oral anticoagulants for preventing stroke in patients with non-valvular atrial fibrillation and no previous history of stroke or transient ischemic attacks. Cochrane Database Syst Rev. 2000;(2):CD001927.

⁵ Taylor FC, Cohen H, Ebrahim S. Systematic review of long term anticoagulation or antiplatelet treatment in patients with non-rheumatic atrial fibrillation. BMJ. 2001 Feb 10;322(7282):321-6.

⁶ Mant J, Hobbs FD, Fletcher K, Roalfe A, Fitzmaurice D, Lip GY, Murray E; BAFTA investigators; Midland Research Practices Network (MidReC). Warfarin versus aspirin for stroke prevention in an

elderly community population with atrial fibrillation (the Birmingham Atrial Fibrillation Treatment of the Aged Study, BAFTA): a randomised controlled trial. *Lancet*. 2007 Aug 11;370(9586):493-503.

subgroup period of 2013-16 in the present study. i) They claim to be especially interested in the period Jan-Mar as if this was prespecified, which apparently it wasn't judging from their previous response. Further there is no statistical testing of the observation that rates are highest in the first 3 months. Also Figure 2 could present the spread of the data for the combined 2008-2016 period.

I do not feel that these comments represent "fatal flaws" but simply represent honest and acceptable scientific differences of opinion.

Our previously published study (BMJ 2018;361:k1717), cited in the current manuscript, provides the full analysis regarding stroke risk in 'resolved AF'. The temporal subgroup analysis revealed that stroke risk did not remain constant over time, but has, in recent years, become similar to risk in patients with ongoing AF, perhaps due to changes in the way the code is used. In specifically highlighting the subgroup analysis for recent years, we wished to draw attention to the *current* risk observed in 'resolved AF', which will be most relevant to clinicians.

The period Jan-Mar was not pre-specified. We simply plotted the monthly frequency of 'resolved AF' coding to explore whether there was any difference to be seen. However, anecdotal observations within the team, who have experience of working with general practitioners, had suggested that general practices may carry out increased assessment and coding related to their QOF assessment (for all QOF conditions, not just AF) shortly before their QOF submission. Visual assessment of the plots and data suggested that Jan-Mar are the months when use of the 'resolved AF' code is at its highest.

As requested, we have added an indication of the spread of data in Figure 2 for the periods where several years were combined (2000-2005 and 2008-2016): we have amended the chart to show median and interquartile range for the annual incidence rates, as below. We thank the reviewer for this suggestion.

UNIVERSITY OF
BIRMINGHAM

REVIEWER	Gaurav A. Upadhyay, MD Center for Arrhythmia Care Heart and Vascular Center Pritzker School of Medicine The University of Chicago Medicine
REVIEW RETURNED	03-Oct-2019

GENERAL COMMENTS	The authors have addressed my questions and the paper makes important observations which would be valuable for the literature. Thank you. The term "resolved AF" may well be a medical misnomer. Patients who have documented AF at one point in their lives may always be at a small risk of subsequent AF. The key observation from the paper which is inferred from the findings but cannot be directly assessed is the motivation with which the "resolved AF" code has been utilized. It may be that the code was utilized significantly more due to "catch up" after changes in QOF requirements and not due to any other more problematic cause. At least the possibility of a benign motivation should be acknowledged in the Discussion. In addition, if there is background on how this code initially entered the medical lexicon, it would be a very helpful context. The reworded Conclusion is also valuable as it advocates uncoupling the anticoagulation assessment from the "resolved AF" use, and may well be a worthwhile goal.
--

VERSION 3 – AUTHOR RESPONSE

Reviewer 2 Comments:

- The authors have addressed my questions and the paper makes important observations which would be valuable for the literature. Thank you.

We thank the reviewer for the positive feedback.

- The term "resolved AF" may well be a medical misnomer. Patients who have documented AF at one point in their lives may always be at a small risk of subsequent AF. The key observation from the paper which is inferred from the findings but cannot be directly assessed is the motivation with which the "resolved AF" code has been utilized. It may be that the code was utilized significantly more due to "catch up" after changes in QOF requirements and not due to any other more problematic cause. At least the possibility of a benign motivation should be acknowledged in the Discussion.

Following the reviewer's previous suggestion, we added the following comment to the Discussion section (first paragraph): 'it is possible that this increase [in incidence of the 'resolved AF' code after QOF] was in part the result of practices 'catching up' with recording 'resolved AF' following the introduction of QOF'.

Throughout the manuscript, we have avoided using causal language to assert a causal relationship between the introduction of QOF and the observed trends in use of the 'resolved AF' code. We have simply described the observed temporal trends alongside the QOF timeline, and, while acknowledging that at least some of the increase in use may be related to 'catch up', have left it for the reader to infer possible explanations.

- In addition, if there is background on how this code initially entered the medical lexicon, it would be a very helpful context.

Unfortunately, the authors are not aware of when or how the 'resolved AF' term first came into use and are unable to identify any previous literature addressing this topic; indeed, the reason for our interest in this subject was the dearth of previous research relating to 'resolved AF'. The corresponding Read code has been in use for many years, prior to the study start date. The Read code, 212R, sits among the Examination/signs ('2...') procedural Read codes; these are sometimes added/requested by clinicians according to need.

The 'AF resolved' code has been used as an exclusion criterion for AF management since the introduction of AF into QOF. We have now clarified this in the manuscript as follows: 'From April 2006, general practices were required to maintain a register of patients with AF and to record whether eligible patients were prescribed anticoagulants or antiplatelets; patients with a code indicating 'resolved AF' are excluded from this register.'

- The reworded Conclusion is also valuable as it advocates uncoupling the anticoagulation assessment from the "resolved AF" use, and may well be a worthwhile goal.